# Breakdown of supersaturation barrier links protein folding to amyloid formation

Masahiro Noji[1,11], Tatsushi Samejima[1], Keiichi Yamaguchi[1,12], Masatomo So[1], Keisuke Yuzu[2], Eri Chatani[2], Yoko Akazawa-Ogawa[3], Yoshihisa Hagihara[3], Yasushi Kawata[4], Kensuke Ikenaka[5], Hideki Mochizuki[5], József Kardos[6], Daniel E. Otzen[7], Vittorio Bellotti[8,9], Johannes Buchner[10] & Yuji Goto[1,12 ✉]

The thermodynamic hypothesis of protein folding, known as the "Anfinsen's dogma" states that the native structure of a protein represents a free energy minimum determined by the amino acid sequence. However, inconsistent with the Anfinsen's dogma, globular proteins can misfold to form amyloid fibrils, which are ordered aggregates associated with diseases such as Alzheimer's and Parkinson's diseases. Here, we present a general concept for the link between folding and misfolding. We tested the accessibility of the amyloid state for various proteins upon heating and agitation. Many of them showed Anfinsen-like reversible unfolding upon heating, but formed amyloid fibrils upon agitation at high temperatures. We show that folding and amyloid formation are separated by the supersaturation barrier of a protein. Its breakdown is required to shift the protein to the amyloid pathway. Thus, the breakdown of supersaturation links the Anfinsen's intramolecular folding universe and the intermolecular misfolding universe.

[1] Institute for Protein Research, Osaka University, Yamadaoka 3-2, Suita, Osaka 565-0871, Japan. [2] Department of Chemistry, Graduate School of Science, Kobe University, Hyogo 657-8501, Japan. [3] National Institute of Advanced Industrial Science and Technology (AIST), 1-8-31 Midorigaoka, Ikeda, Osaka 563-8577, Japan. [4] Department of Chemistry and Biotechnology, Graduate School of Engineering, Tottori University, Tottori 680-8552, Japan. [5] Department of Neurology, Graduate School of Medicine, Osaka University, Yamadaoka 2-2, Suita, Osaka 565-0871, Japan. [6] ELTE NAP Neuroimmunology Research Group, Department of Biochemistry, Eötvös Loránd University, Pázmány P. sétány 1/C, Budapest 1117, Hungary. [7] Interdisciplinary Nanoscience Center (iNANO), Department of Molecular Biology and Genetics, Aarhus University, Gustav Wieds Vej 14, DK-8000 Aarhus C, Denmark. [8] National Amyloidosis Centre, Centre for Amyloidosis and Acute Phase Proteins, University College London, London NW3 2PF, UK. [9] Department of Molecular Medicine, Institute of Biochemistry, University of Pavia, Pavia 27100, Italy. [10] Center for Integrated Protein Science at the Department Chemie, Technische Universität München, Lichtenbergstrasse 4, D-85747 Garching, Germany. [11] Present address: Graduate School of Human and Environmental Studies, Kyoto University, Yoshidanihonmatsucho, Sakyo-ku, Kyoto 606-8316, Japan. [12] Present address: Global Center for Medical Engineering and Informatics, Osaka University, 2-1 Yamadaoka, Suita, Osaka 565-0871, Japan. ✉email: gtyj8126@protein.osaka-u.ac.jp

Many globular proteins can form amyloid fibrils, misfolded ordered aggregates associated with serious amyloidosis[1–5]. Current concepts argue that folding and misfolding are alternative reactions of unfolded proteins[6–8]. However, which principles govern the relationship between folding and misfolding is unknown. Now that the atomic structures for various amyloid fibrils[3–5,9,10] show that they are indeed ordered structures achieved by hydrophobic interactions, hydrogen bonds, and van der Waals interactions, it is important to establish a unifying mechanism explaining both folding/unfolding and amyloid polymerization/depolymerization.

Denatured proteins exist in vivo often at concentrations which are supersaturated concerning solubility[11–14]. Supersaturation is a fundamental phenomenon of nature, determining the phase transition of substances. It is required for formation of crystals and involved in the super-cooling of water prior to ice formation. The same will be true for crystal-like amyloid formation. Here, an additional trigger (e.g., ultrasonic agitation) can disrupt this metastable state leading to amyloid formation[14–16].

To test which factors determine the accessibility of the native, unfolded, and amyloid states for a protein, we focused on the breakdown of supersaturation as a critical factor for a pathway to amyloid fibrils. We heated proteins with or without stirrer agitation and monitored amyloid formation via the amyloid specific thioflavin T (ThT) fluorescence and the total amount of aggregates via light scattering (LS) (Supplementary Fig. 1). The proteins included not only typical amyloidogenic proteins, but also several textbook proteins used previously in folding/unfolding studies[17] (Table 1). Many of them showed Anfinsen-like reversible unfolding/refolding upon heating, but formed amyloid fibrils with or without agitation at high temperatures. This behavior is explained by the persistence of supersaturation, which depends on the flexibility of denatured states.

## Results

**Proteins with an immunoglobulin fold**. We first examined proteins with an immunoglobulin fold-like $\beta_2$-microglobulin ($\beta$2m), a blood protein responsible for dialysis-related amyloidosis: the variable ($V_L$)[3,18] and constant ($C_L$)[19] domains of the immunoglobulin light chain. An excess amount of specific variants of monoclonal light chains or $V_L$ fragments secreted into the blood stream forms amyloids causing AL amyloidosis, whereas the $C_L$ domain is assumed to modulate $V_L$'s amyloidogenicity[3,18,20]. At pH 7.0, upon heating under stirring, $V_L$ (PAT-1) exhibited an increase in ThT fluorescence at approximately 65 °C, whereas no reaction was observed without stirring (Fig. 1a and Supplementary Fig. 2a). Upon cooling, the ThT fluorescence increased linearly which is an intrinsic property of ThT[15,16,21]. Amyloid formation was confirmed by circular dichroism (CD) spectroscopy and transmission electron microscopy (TEM) (Fig. 1b and Supplementary Fig. 2b). These results were consistent with those observed for $\beta$2m[16].

$C_L$ also demonstrated heating-induced amyloid formation (Supplementary Fig. 3). Although fibrous aggregates were observed by TEM, the conformational change monitored by CD was small compared with that of $V_L$ (Supplementary Fig. 3). These results were consistent with the previously reported high and low amyloidogenicity of $V_L$[18] and $C_L$[19], respectively.

**Small globular proteins**. We also examined various small globular proteins with tightly folded native structure. Hen egg white lysozyme (HEWL) is one of the most extensively studied proteins in terms of protein unfolding/folding and it is well known that HEWL forms amyloid fibrils at high temperatures under acidic conditions[22,23]. Our methodology revealed that stirrer agitation

## Table 1 Target proteins/peptides used in this study.

| General properties | | | | | | | | Experimental conditions | | | | Transition type |
|---|---|---|---|---|---|---|---|---|---|---|---|---|
| Class | Name | Amino acid residues | Mw (kDa) | pI | $\Delta S_{conf}{}^a$ (J K$^{-1}$ mol$^{-1}$) Main chain | S-S bonds | Ave. hydrophobicity$^b$ | Amyloidosis | Protein (mg/ml) | (µM) | pH | NaCl (M) | |
| Short peptides | K3 | 22 | 2.5 | 4.5 | 462 | 0 | 0.57 | - | 0.06 | 25 | 2.0 | 0.1 | |
| | pOVA | 23 | 2.5 | 5.8 | 483 | 0 | 0.87 | - | 0.1 | 41 | 8.0 | 0.1 | |
| | Glucagon | 29 | 3.5 | 6.8 | 609 | 0 | 0.37 | - | 0.2 | 57 | 7.0 | 0.1 | |
| | IAPP | 37 | 3.9 | 8.9 | 777 | -29 | 0.29 | Type II diabetes | 0.1 | 26 | 7.0 | 0.1 | A |
| | Aβ40 | 40 | 4.3 | 5.3 | 840 | 0 | 0.53 | Alzheimer's disease | 0.1 | 23 | 7.0 | 0.1 | |
| | Insulin$^c$ | 51 | 5.8 | 5.4 | - | - | - | Insulin-derived amyloidosis | 0.2 | 35 | 7.0 | 0.1 | |
| Immunoglobulin folds | β2m | 99 | 12 | 6.1 | 2079 | -59 | 0.32 | DRA | 0.1 | 8.5 | 7.0 | 0.0–3.0 | S |
| | $C_L$ | 106 | 11 | 5.6 | 2226 | -60 | 0.42 | AL amyloidosis | 0.2 | 18 | 7.0 | 0.5 | |
| | $V_L$ (PAT) | 112 | 12 | 5.0 | 2352 | -61 | 0.44 | | 0.1 | 8.5 | 7.0 | 0.2 | |
| Globular proteins | Ubiquitin | 76 | 8.6 | 6.6 | 1596 | 0 | 0.22 | - | 0.2 | 23 | 2.0 | 0.5 | |
| | αLA | 123 | 14 | 4.8 | 2583 | -220 | 0.23 | - | 0.2 | 14 | 2.0 | 0.5 | |
| | RNaseA | 124 | 14 | 8.6 | 2604 | -209. | 0.18 | - | 0.2 | 15 | 5.0 | 1.0 | |
| | TTR | 127 | 14 | 5.3 | 2667 | 0 | 0.23 | SSA, FAP, FAC | 0.2 | 14 | 2.0 | 0.1 | |
| | HEWL | 129 | 14 | 9.3 | 2709 | -221 | 0.11 | - | 0.2 | 14 | 2.0 | 0.5 | |
| | OVA | 386 | 43 | 5.2 | 8106 | -57 | 0.33 | - | 2.0 | 47 | 8.0 | 0.1 | |
| IDPs | αSyn | 140 | 14 | 4.7 | 2940 | 0 | 0.20 | Parkinson's disease | 0.2 | 14 | 7.0 | 1.0 | B |
| | TDP-43 | 414 | 45 | 5.8 | 8694 | 0 | 0.18 | ALS, FTD | 0.2 | 4.5 | 7.0 | 1.0 | S |
| | Tau | 441 | 46 | 8.2 | 9261 | 0 | 0.02 | Alzheimer's disease | 0.4 | 8.7 | 7.0 | 0.15 | B |

$^a$Main chain $\Delta S_{conf}$ was estimated assuming that the main chain contributes 21 J K$^{-1}$ per mole of residues[49]. Reduction of $\Delta S_{conf}$ by S-S bonds was estimated by Pace et al[50].
$^b$Average hydrophobicity estimated by CamSol[44].
$^c$The vacant cells of insulin were not estimated because of the complexity caused by two constituent chains.

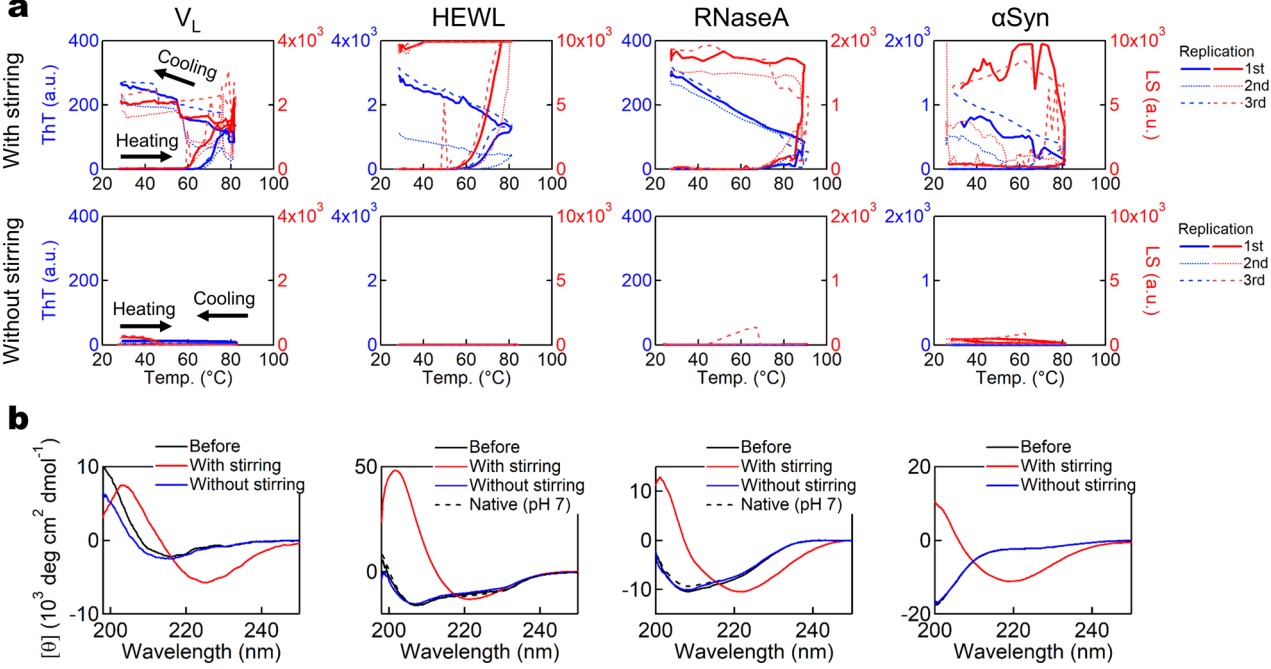

**Fig. 1 Heating- and agitation-dependent amyloid formation of type S proteins. a** ThT assays of $V_L$, HEWL, RNaseA, and αSyn upon heating in the presence (upper) or absence (lower) of stirring. The intensities of ThT fluorescence and LS are indicated by blue and red lines, respectively. $n = 3$ and triplicate runs are shown with helper arrows. **b** CD spectra of each sample before and after the ThT assays. The line representations are described in the panels.

was required for amyloid formation of HEWL at pH 2 (Fig. 1 and Supplementary Fig. 2). Based on differential scanning calorimetry (DSC) measurements, heating-induced amyloid fibrils of HEWL had seeding activity, as previously observed for β2m[16] (Supplementary Fig. 4a, b). Seeding activity is a defining amyloid property, which reflects the crystal-like mechanism of amyloid formation[11,12].

Transthyretin (TTR) is a tetrameric transporter protein in the plasma, which has been studied intensively because of its association with fatal amyloidoses such as senile systemic amyloidosis and familial amyloid polyneuropathy[3,9]. In addition to a large number of pathogenic familial mutants, wild-type TTR can form amyloid fibrils in patients[24]. However, spontaneous amyloid formation of wild-type TTR in vitro is difficult unless fragmentation is induced by proteolysis[25]. We found that wild-type full length TTR exhibited amyloid formation upon incubation under stirring at pH 2.0 and 50 °C, whereas TTR retained a native-like conformation without stirring (Fig. 2a–c). This reaction occurs in a narrow range of NaCl concentrations, from 50–150 mM (Fig. 2d). It has been reported that TTR readily forms ThT-negative and seeding-incompetent filamentous aggregates[26] with a curvilinear morphology[24]. In contrast, the amyloid fibrils we prepared at pH 2.0 and at 50 °C had seeding activity under the same solvent conditions (Fig. 2e).

We also investigated the behavior of ribonuclease A (RNaseA) and α-lactalbumin (αLA), two model proteins used in many folding studies[17,27]. The hinge loop-expanded mutant of RNaseA was reported to generate amyloid-like fibrils via 3D domain swapping, whereas the wild-type RNaseA did not[28,29]. Most interestingly, we successfully induced amyloid formation of wild-type RNaseA with 1.0 M NaCl at pH 5.0, for which stirring was essential (Fig. 1 and Supplementary Fig. 2). Seeding activity was also confirmed by DSC measurements (Supplementary Fig. 4c). For αLA, it has been reported that amyloid fibrils can be formed at low pH or upon reduction of disulfide bonds[30]. In our experiments, intact αLA produced amyloid fibrils upon heating

under agitation, consistent with previous reports, although stirring was required in our case (Supplementary Fig. 5).

Ubiquitin is a small protein which tags other proteins for degradation[31]. It shows reversible unfolding[17,32]. Recently, amyloid-like fibril formation of heat-treated poly-ubiquitin chains was reported[33]. We examined whether mono-ubiquitin can form amyloid fibrils by heating under agitation. An increase in ThT fluorescence was noted upon incubation at 90 °C and pH 2, for which stirring was essential (Supplementary Fig. 6). Of note, the ThT intensity was weak despite strong LS even after cooling to 25 °C. The CD spectra reflected a β-rich conformation and fibrous aggregates were observed by TEM.

**Highly amyloidogenic short peptides**. We then examined amyloidogenic short peptides without ordered structures. Amyloid β peptides 1–40 (Aβ40) or 1–42 (Aβ42) are two of the most well-known amyloidogenic peptides[3,10]. Amyloid formations of Aβ40 and Aβ42 have been suggested to play key roles in Alzheimer's disease, in which senile plaques in the brain contain a large amount of Aβ40 and Aβ42 fibrils. In our experiments, Aβ40 formed amyloid fibrils at pH 7.0 under agitation monitored by ThT fluorescence, CD, and TEM (Fig. 3 and Supplementary Fig. 7). In contrast, without agitation, no amyloid formation was observed by ThT fluorescence or CD. However, even without agitation, the heat treatment and subsequent overnight incubation caused amyloid formation according to TEM (Supplementary Fig. 7b).

We obtained similar results for the 29-residue peptide hormone glucagon[34]. Although no increase in ThT fluorescence was observed without agitation during the heating experiments, fibrous aggregates were observed by TEM upon overnight incubation after the heating experiments (Fig. 3 and Supplementary Fig. 7).

Islet amyloid polypeptide (IAPP), also known as amylin, is a 37-residue peptide hormone and the amyloid deposition of IAPP in pancreatic β-cells may cause type II diabetes[3]. Upon heating,

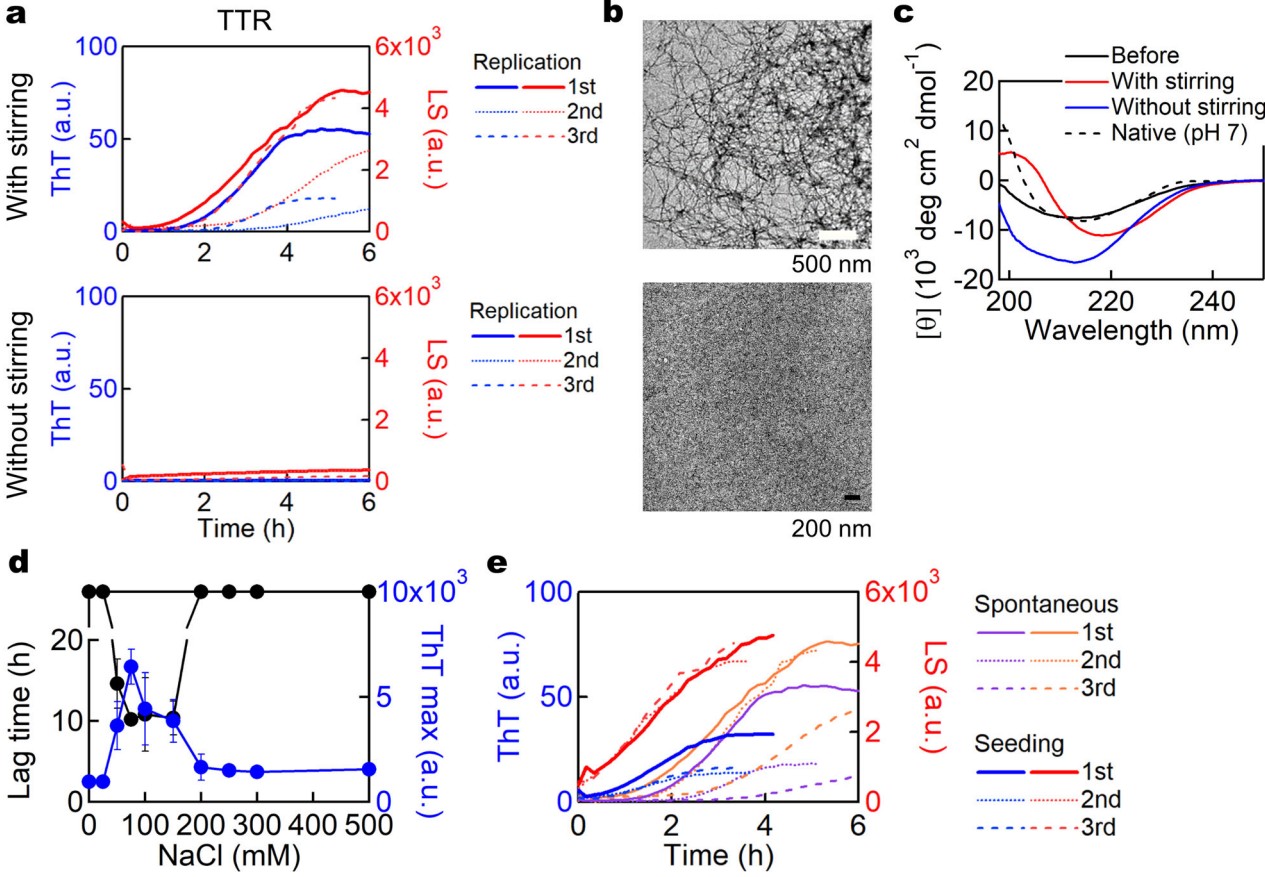

**Fig. 2 Amyloid formation of wild-type TTR. a** ThT assays of TTR upon incubation in the presence (upper) or absence (lower) of stirring. $n = 3$. **b** TEM images of the samples after the ThT assay. Scale bar; 500 nm. **c** CD spectra of the samples before and after the ThT assays. **d** Dependence of TTR amyloid formation on NaCl concentration. Lag times and ThT maximal intensities are indicated by black and blue lines, respectively. $n = 5$. **e** ThT assays with (solid lines) or without (dashed lines) 5% seeds under the same conditions as (**a**). $n = 3$.

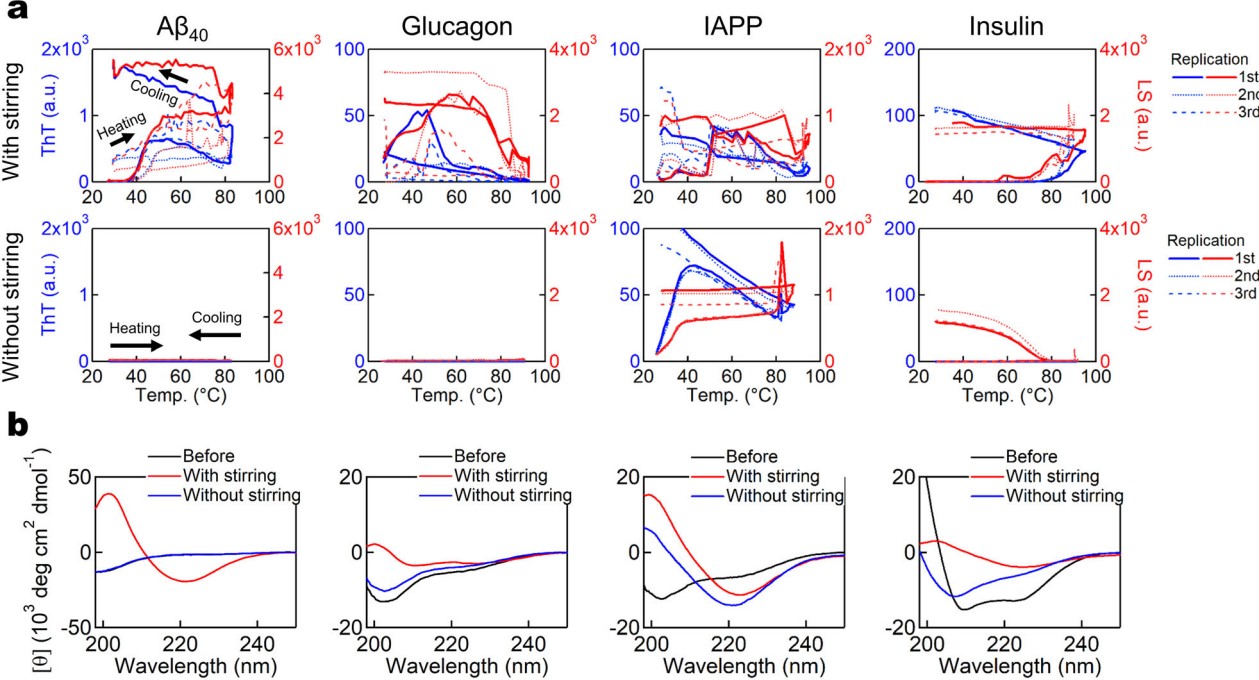

**Fig. 3 Effects of heating and agitation on type A proteins. a** ThT assays of $A\beta_{40}$, glucagon, IAPP, and insulin upon heating in the presence (upper) or absence (lower) of stirring. $n = 3$. **b** CD spectra of each sample before and after the ThT assays.

IAPP exhibited amyloid formation both in the presence and absence of stirring at pH 7, which was confirmed by ThT fluorescence, CD, and TEM (Fig. 3 and Supplementary Fig. 7), suggesting that the supersaturation barrier is not high.

Insulin, one of the peptide hormones secreted by the pancreatic islets in addition to IAPP, is composed of a 21-residue A chain and a 30-residue B chain, which are linked by two disulfide bonds. Insulin forms amyloid fibrils upon heating and the degree of agitation was reported to affect the fibril morphology[3,35]. Under agitation, ThT fluorescence increased at ~80 °C (Fig. 3a and Supplementary Fig. 7). Without stirring, no increase in fluorescence was observed upon heating. Upon cooling, LS increased markedly, suggesting the formation of amorphous aggregates (Fig. 3a and Supplementary Fig. 7). It seems that insulin entered an alternative pathway leading to amorphous aggregation upon heating and cooling under quiescence.

**Relatively large IDPs.** Lastly, we examined the heating-induced amyloid formation of several intrinsically disordered proteins (IDPs). IDPs are excluded from Anfinsen's dogma because they lack ordered native structures, even if the ensemble of non-native structures is still determined by sequence. One of the most amyloidogenic IDPs is α-synuclein (αSyn). Amyloid formation of αSyn is highly associated with three types of synucleinopathies: Parkinson's disease, dementia with Lewy bodies, and multiple system atrophy[3,36]. According to the ThT assays under stirring at pH 7.0 with 1.0 M NaCl, heating-induced amyloid formation of αSyn occurred readily, starting at ~60 °C, whereas it did not occur without stirring (Fig. 1a and Supplementary Fig. 2a). CD and TEM exhibited amyloid formation only under stirring (Fig. 1b and Supplementary Fig. 2b). To gain more insight into the fibril formation of αSyn, we carried out detailed experiments examining the dependence of αSyn amyloid formation on the salt concentration and temperature. We used sodium sulfate and ultrasonic agitation to enhance the salt[23] and agitation[14,37] effects, respectively. Amyloid formation was promoted at high temperatures and relatively high concentrations of sodium sulfate without amorphous aggregation (Supplementary Fig. 8). Therefore, even αSyn, a classical IDP, followed the pattern of strict dependence on heating and agitation.

To examine whether the observed effects for αSyn are typical for IDPs, we studied two further family members. The microtubule-associated protein tau[3], an IDP comprising 441 amino acid residues, forms tau filamentous aggregates which are the main component of neurofibrillary tangles in the brain of Alzheimer's disease patients, suggesting that tau and Aβ work synergistically to cause the disease. TAR DNA-binding protein 43 (TDP-43)[3], a 414-residue nuclear IDP playing a central role in RNA metabolism, forms aggregates in patients with amyotrophic lateral sclerosis and frontotemporal dementia.

Using tau and TDP-43, we examined the effects of heating in the presence or absence of stirring. For both proteins, ThT assays suggested that the heating-induced amyloid formation was independent of stirring (Fig. 4a), although no apparent conformational change was observed by CD (Fig. 4b). Both exhibited relatively strong ThT fluorescence signals, however, when examined by TEM, only tau exhibited clear fibrous morphology (Supplementary Fig. 9b). These results were different from those for αSyn or amyloidogenic globular proteins, suggesting that amorphous aggregates dominated, although they may contain amyloid core regions like in the case of ovalbumin (OVA)[21].

**Protein concentration dependence.** Because protein aggregation occurs above the solubility limit, the aggregation behavior is likely to be dependent on the protein concentration. We performed the heating experiments with β2m at 0.1, 1.0, 2.0, and 10.0 mg/ml β2m at pH 7.0 (Supplementary Fig. 10). With stirring, although the kinetics of amyloid formation were similar at 0.1, 1.0, and 2.0 mg/ml β2m, it was slightly accelerated at 10 mg/ml β2m. The results suggested that, at low β2m concentrations, the rate-limiting step was not an aggregation of monomers but a process related to unfolding of the native state. At 0.1 mg/ml β2m without stirring, nothing happened upon heating. However, at 10 mg/ml β2m without stirring, amorphous aggregation occurred at approximately 60 °C, indicating that the aggregation behavior changed from the supersaturation-limited amyloid formation to the spontaneous amorphous aggregation.

## Discussion

Our study on the formation of amyloid and amorphous aggregates states of proteins revealed principles that determine the shift in pathways. According to their aggregation behavior, three types of proteins can be defined (Table 1 and Supplementary Table 1).

**Type S proteins.** The first type of proteins shows a strict dependence on agitation for amyloid formation at high temperatures. We call this transition the "strictly supersaturation-dependent transition" or "S transition". Proteins exhibiting S transitions include those with a native conformation (β2m, $V_L$, $C_L$, HEWL, and RNaseA) and αSyn. TTR can also be included in Type S. Although the linkage of unfolding and amyloid polymerization destabilizes the native state[16] (Supplementary Fig. 11), this is not sufficient for amyloid formation as it is strictly dependent on agitation or seeding.

**Type A proteins.** The second type of proteins exhibits spontaneous amyloid formation at high temperatures even without agitation. We refer to this type of transition as "autonomous amyloid-forming transition" or "A transition". Type A proteins include insulin, glucagon, IAPP, and Aβ40. In other words, the high amyloidogenicity of these relatively short amyloid peptides do not exhibit intrinsic barriers preventing amyloid formation.

**Type B proteins.** The third type of proteins often produces amorphous aggregates at high temperatures without a lag phase. We call this transition the "boiled egg-like transition" or "B transition". Type B proteins (i.e., TDP-43, tau, and OVA) are relatively large and it is possible that their overall amorphous characteristics include amyloid cores (β-spines) producing a "fuzzy coat" morphology[38].

We found that three types (S, A, and B) of transitions with distinct responses to heating can be located on a general aggregation phase diagram dependent on the driving forces of precipitation and protein solubility[14,16,37] (Fig. 5a). The S, A, and B transitions are indicated by green, orange, and purple arrows, respectively. This type of diagram is often used to illustrate the crystallization and amorphous precipitation of native proteins and, moreover, for solutes in general[39]. The driving force of precipitation is influenced by various parameters such as salt concentration or temperature. Hydrophobic interactions and conformational entropies, which respectively favor and disfavor protein interactions, both increase with temperature[17]. Therefore, amyloid formation and protein folding exhibit cold- and heat-denaturation phenomena[7,15,40,41]. We previously studied with β2m at pH 7.0 the relationship between the temperature dependence of solubility and amyloid formation, showing both of lowest solubility and maximal amyloid formation occurred at ~70 °C[16]. In this study, only heating-dependent promotion of

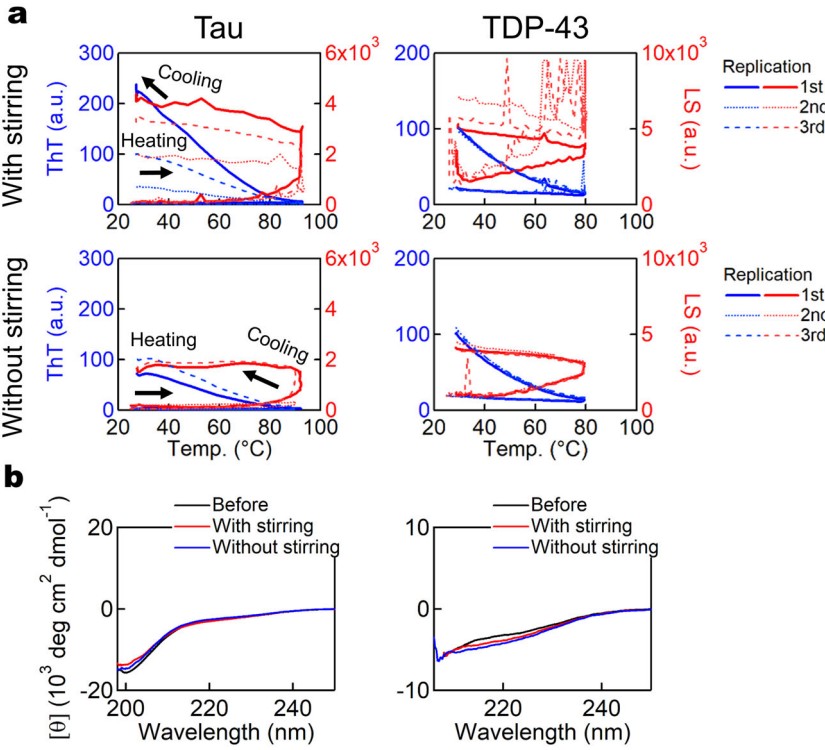

**Fig. 4 Boiled egg-like aggregation of type B proteins. a** ThT assays of tau and TDP-43 upon heating in the presence (upper) or absence (lower) of stirring. $n = 3$. **b** CD spectra of each sample before and after the ThT assays.

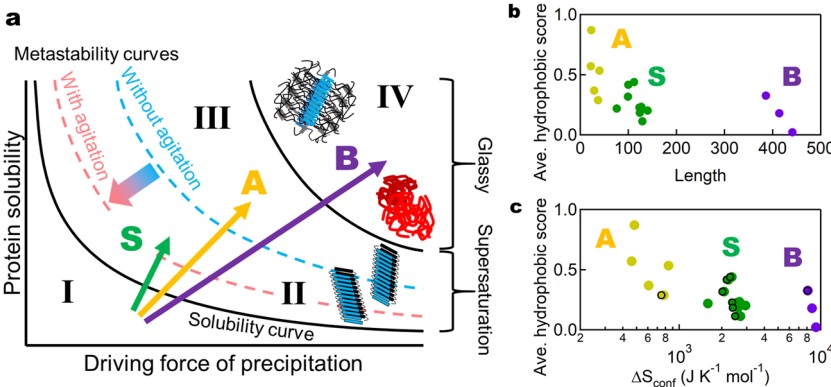

**Fig. 5 General schematic conformational phase diagram and the three transitions.** Three types of amyloidogenic proteins were plotted on a general phase diagram of aggregation (**a**) and diagrams of average hydrophobicity vs. number of amino acid residues (**b**) or $\Delta S_{conf}$ (**c**). In **a**, an increase in the salt concentration or temperature often increases driving force of precipitation (abscissa), thus decreasing protein solubility (ordinate). In **c**, $\Delta S_{conf}$ represents an increase upon denaturation of the main-chain with (points with frame) and without (points without frame) the contribution of disulfide bonds (Table 1).

amyloid formation was focused on, where hydrophobic interactions increase with an increase in temperature.

The general phase diagram dependent on temperature consists of a soluble region (Region I), metastable region (Region II), labile region (Region III), and glassy region (Region IV)[15,16,37,39]. Below solubility (Region I), monomers are thermodynamically stable. In the metastable region (Region II), supersaturation is assumed to persist in the absence of seeding or agitation. In the labile region (Region III), spontaneous nucleation occurs after a certain lag time. Finally, the glassy region (Region IV) is dominated by amorphous aggregation occurring without a lag time possibly by concomitant formation of many nuclei.

Thus, the S, A, and B transitions represent those from below solubility (Region I) to metastable region (Region II), labile region (Region III), and glassy region (Region IV), respectively. These

indicate that, with an increase in driving force of precipitation at high temperatures, the aggregation behavior followed exactly as expected for solutes in general[39].

In terms of the phase diagram of conformational states (Fig. 5a), stirring or ultrasonication is a kinetic factor modifying the apparent phase diagram. It is likely that the boundary between the metastable and labile regions is shifted downward upon agitation, decreasing the barrier of supersaturation and inducing spontaneous amyloid formation[37]. At a molecular level, we proposed a theoretical model of ultrasonication-induced amyloid formation[42], where denatured monomers are captured on the bubble surface during its growth and highly condensed by subsequent bubble collapse, so that they are transiently exposed to high temperatures. Thus, the dual effects of local condensation and local heating contribute to dramatically enhance the

nucleation reaction. We further suggested that the local condensation of the denatured proteins at the water–air interface forms seed-competent conformation[23]. Consistent with this, marked suppression of amyloid nucleation of Aβ was shown under agitation without water-air interface[43].

To address the mechanism underlying the distinct amyloidogenic transitions, we examined the relationship between transition types (i.e., S, A and B types) and various factors, which might determine the types. Although determining the solubility of denatured proteins experimentally at high temperatures will be important, it is impractical at this stage. Thus, we examined factors related to solubility although they are relative values estimated at ambient temperatures. These include average hydrophobicity score[44], CamSol score[44], protein solubility score[45], AGGRESCAN score[46], and Tango AGG score[46] as well as net charge (Supplementary Table 1). For the proteins and peptides examined, the scores of various factors were plotted against the number of amino acid residues (Fig. 5b, Supplementary Fig. 12). The plots included our results using short amyloidogenic peptides obtained from full-length proteins, i.e. K3[47] and OVA peptide (pOVA)[21].

It is evident that the total residue number (abscissa) is the most dominant factor determining the different types. Then, hydrophobic score or AGGRESCAN score (Supplementary Fig. 12) (ordinate) showed notable correlation with distinct amyloid types. On the other hand, CamSol score, protein solubility score, Tango AGG score, and net charge did not distinguish the types. When viewed from the size and hydrophobicity (Fig. 5b), the S proteins had a moderate size and moderate hydrophobicity, the A proteins had a short length and high hydrophobicity, and the B proteins had a long length and low hydrophobicity. Once short peptides with high hydrophobicity were prepared from S or B proteins, they became the A type (e.g., K3[47] and pOVA[21]).

Although intrinsic properties such as the total residue number and hydrophobicity are important factors to determine the types, it is also obvious that the transition type depends on the solvent conditions. When the driving forces of precipitation are increased for a particular protein, the transition type changes from S to A and B, as for example, the amyloidogenic transition exhibited by acid-denatured β2m[15,16]. Here, changes from S to A and B was observed upon increasing the salt concentration. Although TTR exhibited the S-type transition in this study, it also showed a B-type transition with a curvilinear morphology[24,26], possibly because the driving forces were excessive. The driving forces also increase with an increase in protein concentration; therefore, the type of transition will change from S to A and B at higher protein concentrations. This change of transition type occurred for β2m at pH 7.0, where the S-type transition at 0.1 mg/ml changed to the B-type transition at 10 mg/ml (Supplementary Fig. 10). This corresponds to a move along the x-axis in the phase diagram (Fig. 5a) and therefore automatically leads to a phase border crossing.

Another important factor is the disulfide bond[47,48]. The reduction of disulfide bonds often reduces the amyloidogenicity, as demonstrated for β2m: the S-type transition under acidic conditions changed to B-type. These roles of disulfide bonds suggested that a more appropriate scale for evaluating the different types of amyloidogenic proteins is "conformational flexibility of the denatured state".

Although the estimation of conformational entropy in denatured state as well as that of disulfide bonds is not straightforward, we tentatively employed the methodology often used for the analysis of conformational stability of globular proteins[17,49,50]. Assuming that the main chain contributes $21\,J\,K^{-1}$ per mole of residues[49], we estimated the intrinsic conformational entropies of

the denatured state "without" disulfide bonds ($\Delta S_{conf}$), which were plotted against various possible factors (Table 1, Fig. 5c, Supplementary Table 1, Supplementary Fig. 13). Again, hydrophobicity and AGGRESCAN scores were factors highly correlating with transition types, suggesting the importance of hydrophobicity-related interactions.

Although the effects of disulfide bonds in reducing the conformational entropy have been addressed[17,50], they are in fact small in comparison with intrinsic $\Delta S_{conf}$ (Table 1, Fig. 5c, Supplementary Fig. 13). More importantly, the disulfide bonds stabilize hydrophobic cores that persist in the denatured state and thus increase amyloidogenicity, as demonstrated for acid-denatured β2m[47,48]. Taken together, synergetic effects of disulfide bonds (i.e., decreasing the intrinsic conformational entropy and stabilizing the hydrophobic cores, the latter not included in Fig. 5c) lead to the large decrease in flexibility of the denatured states.

Finally, one of the most important phenomena related to amyloid formation is liquid–liquid phase separation observed increasingly in disordered proteins[51,52]. There are cases that the amyloid formation is preceded by the liquid–liquid phase separation. For an example, low-complexity domain of FUS protein formed the phase-separated droplets before formation of more stable amyloid fibrils[53]. The results are consistent with the Ostwald's ripening rule of crystallization, in which morphologies of crystals change with time guided by their kinetic accessibilities and thermodynamic stabilities[14,23,54]. We assume that the "macroscopic" phase diagram of conformational states as Fig. 5a will be also useful for understanding the liquid–liquid phase separation, where "microscopic" phase diagram might apply to each droplet system.

For β2m, a type S protein, a three-state mechanism has been proposed[16] where both folding and misfolding purely depend on the Gibbs free energy change (Supplementary Fig. 11 and Supplementary Movie 1). Thus, we can apply this in general to S type proteins. In this concept, before the breakdown of supersaturation, a "protein concentration-independent" two-state mechanism applies. Upon the breakdown of supersaturation, a three-state mechanism between the native, unfolded, and "protein concentration-dependent" amyloid states determines the overall equilibrium. The transition from the two-state mechanism to the three-state mechanism shifts the overall equilibrium to the direction of amyloid fibrils, apparently destabilizing the native state by the law of mass action[16].

In conclusion, our results provide another example that the effects of extreme conditions of temperature or pressure on folding/misfolding are important for clarifying the phenomena under physiological conditions[7,51,55,56]. Combined effects of thermodynamics (i.e., reversible unfolding) and kinetics (i.e., breaking supersaturation) have profound implications in deviations from Anfinsen's dogma and possible onset of amyloidoses. The seemingly disparate perturbations (i.e., reversible unfolding, aggitation and breakdown of supersaturation) act cumulatively each other, which seems common to various proteins. The linkage of folding/misfolding and the law of mass action enable amyloid formation even under physiological conditions where only a low amount of unfolded precursor exists. Although the validity of Anfinsen's dogma is often questioned under high protein concentrations where intermolecular interactions are favored[8], the persistence of supersaturation and the difficulty of amyloid formation under quiescent conditions have excluded to address the question experimentally. Our considerations indicates that specifically the breakdown of the kinetic barriers of supersaturation links the Anfinsen's intramolecular folding universe with the "outer" intermolecular misfolding universe.

## Methods

### Proteins and chemicals

*αSyn*. Recombinant human αSyn gene was amplified from cDNA of human brain (Cap site cDNA dT: Nippon gene) by PCR, subcloned into the NdeI and XhoI multicloning site of the expression vector pET23a(+) (Novagen), and expressed in *Escherichia coli* BLR(DE3) (Novagen)[57]. Cells were suspended in purification buffer (50 mM Tris-HCl, pH 7.5, containing 1 mM EDTA, 0.1 mM dithiothreitol (DTT), and 0.1 mM phenylmethylsulfonyl fluoride), disrupted using sonication, and centrifuged (14,000 rpm, 30 min). Streptomycin sulfate (final 2.5%) was added to the supernatant to remove nucleic acids. After removal of nucleic acids by centrifugation, the supernatant was heated to 90 °C for 15 min and then centrifuged. In this step, αSyn remained in the supernatant. The supernatant was precipitated by the addition of solid ammonium sulfate to 70% saturation, centrifuged, and dialyzed overnight and then applied onto a Resource-Q column (Amersham Biosciences) with 50 mM Tris-HCl buffer, pH 7.5, containing 0.1 mM DTT and 0.1 mM phenylmethylsulfonyl fluoride as running buffer. Samples were eluted with a linear gradient of 0–1 M NaCl. Protein concentration of αSyn was determined by using a molar absorption coefficient reported[58].

$V_L$. Plasmid encoding human $V_L$(PAT-1)(1–112) was generated by QuikChange mutagenesis PCR (Agilent, Santa Clara, CA, USA) according to the manufacturer's protocol[18,20]. The plasmid was transformed into *E. coli* BL21 (DE3)-star cells. Protein expression at 37 °C was induced with 1 mM IPTG at an OD600 of 0.6–0.8. Cells were harvested after overnight protein expression, and inclusion bodies were recovered. The pellet was solubilized and unfolded in 25 mM Tris-HCl (pH 8.0), 5 mM ethylenediaminetetraacetic acid (EDTA), 8 M urea and 2mM β-mercaptoethanol at room temperature for at least 2 h. The solubilized protein was injected on a Q-Sepharose column equilibrated in 25 mM Tris-HCl (pH 8.0), 5 mM EDTA and 5 M urea. The proteins were all eluted in the flow-through fraction and subsequently refolded by dialysis against 250 mM Tris-HCl (pH 8.0), 100 mM L-Arg, 5 mM EDTA, 1 mM oxidized glutathione and 0.5 mM reduced glutathione at 4 °C overnight. To remove aggregates and remaining impurities, the proteins were purified using a Superdex 75 16/60 column (GE Healthcare, Uppsala, Sweden) equilibrated in PBS buffer. The recovery and purity of intact protein was verified by SDS-PAGE.

$C_L$. Isolated DNA encoding human type κ $C_L$(1–106) domain was cloned into an *E. coli* expression vector, pAED4[19] and expressed in BL21 (DE3) pLysS cells (Agilent Technologies, La Jolla, CA). The produced protein was accumulated in inclusion bodies. Inclusion body from 1.6 L culture was resuspended in 3 ml of 100 mM Tris-HCl (pH 8.5) containing 10 mM DTT and solubilized by the addition of 0.6 g of solid guanidine hydrochloride (Gdn-HCl). The crude $C_L$ was purified over a Superdex 75 column (GE Healthcare, Waukesha, WI) pre-equilibrated with 6 M urea in 10 mM Tris-HCl (pH 8.5), and then reduced with 20 mM DTT, followed by dialysis against 10 mM Tris-HCl (pH 8.5) with 6 M urea at 4 °C for air oxidation for >48 h. The oxidized sample was dialyzed against 10 mM Na-acetate (pH 4.7) and purified using a Resource S cation-exchange column (GE Healthcare) equilibrated with 10 mM Na-acetate (pH 4.7). The concentration of $C_L$ was determined using a molar extinction coefficient of 10,950 $M^{-1}$ $cm^{-1}$ at 280 nm, respectively, which was determined on the basis of amino acid composition[59].

*TTR*. Human wild-type TTR was expressed using a pET3a vector containing the full-length cDNA for human TTR in *E. coli* BL21(DE3)[60]. Expression colonies were grown to an optical density of 0.6 at 600 nm in Luria-Bertani broth containing 100 μg ml$^{-1}$ ampicillin at 37 °C; protein synthesis was induced with 0.8 mM isopropyl-β-D-thiogalactoside at 16 °C overnight. The following day cells were harvested by centrifugation at 2150 × g for 30 min, the pellet was suspended in buffer containing 25 mM Tris-HCl, 2 mM EDTA, 0.1% Triton, pH 7.4 and sonicated for ten cycles (1 min on/1 min off). The intracellular proteins were fractionated by two cycles of ammonium sulfate precipitation. TTR, which precipitated between 30 and 60% ammonium sulfate, was dissolved in 50 mM Tris-HCl, pH 7.5. The supernatant was heated to 65 °C for 50 min. After removal of pellet by centrifugation, the supernatant was applied onto a Resource-Q column (GE Healthcare Life Science) equilibrated and eluted with a linear gradient of 0.1–0.35 M NaCl. TTR-enriched fractions were dialyzed overnight water at 4 °C and then lyophilized. Purity was confirmed by SDS-PAGE and electrospray ionization mass spectrometry.

*Glucagon*. Pharmaceutical grade human glucagon expressed in *E. coli* and purified (> 98.9%) by Novo Nordisk A/S (Gentofte, Denmark) was kindly given and used[34].

*TDP-43*. The plasmid of human N-terminal His-Tag TDP-43 was kindly gifted from Prof. Yun-Ru Chen (Academia Sinica, Taipei, Taiwan)[61]. The N-terminal His-tagged TDP-43 was transformed and overexpressed in *E. coli* BL21(DE3) (Novagen). The cells were harvested and lysed by BugBuster (Merk) including benzonase nuclease, 0.1 mM phenylmethylsulfonyl fluoride, 0.2 mg/ml lysozyme. TDP-43 accumulated in inclusion bodies. After removal of supernatant by centrifugation, the pellet was dissolved in 50 mM Tris-HCl (pH 8.0)

containing 8.0 M urea, 1 mM DTT. The sample was subjected to a HisTrap FF (GE Healthcare) equilibrated with 8.0 M urea, 50 mM Tris-HCl (pH 8.0), and then proteins were eluted with 500 mM imidazole. The fraction containing the major peak was dialyzed against 25 mM Tris-HCl (pH 8.0), 4.0 M urea, 1 mM DTT and then subjected to a Resource-Q column (Amersham Biosciences) equilibrated with 25 mM Tris-HCl (pH 8.0), 4.0 M urea. The proteins were eluted with a linear gradient of NaCl (0–1.0 M). The fraction containing the major peak was dialyzed against 5 mM Tris-HCl (pH 9.0). Purified His-tagged TDP-43 protein was run on SDS-PAGE and identified by Coomassie blue staining. The recombinant TDP-43 contained extra N-terminal residues MGSSHHHHHHSSGLVPR GSHMLE. Protein concentration was quantified after background subtraction by absorption at 280 nm with the extinction coefficient of 44,380 $M^{-1}$ $cm^{-1}$ according to the equation described by Pace et al.[62]

*Tau40*. A pET23a-hTau40 gene was constructed by ligation of a synthesized hTau40 gene optimized for *E. coli* expression (Thermo Fisher, MA, USA) with a DNA fragment obtained from the expression vector pET-23a(+). Human Tau40 was expressed in *E. coli* BL21(DE3) (Nippon gene Co., Ltd, Tokyo, Japan) transformed with the pET23a-hTau40. Cells were disrupted by ultrasonication in purification buffer (50 mM Tris-HCl, pH 7.4, containing 2mM EDTA, 2mM DTT, and 0.2 mM PMSF) and centrifuged at 22,000 × g for 20 min. NaCl (final 200 mM) was added to the supernatant and heated at 85 °C for 10 min. After centrifugation (22,000 × g, 20 min), streptomycin sulfate (final 2.5%) was added to the supernatant and centrifuged at 22,000 × g for 20 min. Supernatants were dialyzed overnight and applied onto a CM sepharose column (GE Healthcare Life Sciences, Buckinghamshire, UK) with 50 mM Tris-HCl containing 2mM EDTA and 2mM DTT. Samples were eluted with a linear gradient of 0–0.5 M NaCl. The sample solution was dialyzed by 0.05 mM HCl and lyophilized. Molecular weight of hTau40 was assessed by SDS-PAGE and ESI-MS.

*Commercial products*. Bovine αLA, bovine pancreatic RNaseA, and human insulin were purchased from Sigma-Aldrich Co., LLC. (St. Louis, USA). Human ubiquitin was purchased from Funakoshi Co., Ltd. (Tokyo, Japan). Chemically synthesized human Aβ40 and human IAPP were purchased from Peptide Institute, Inc. (Osaka, Japan). All other reagents, including HEWL, were obtained from Nacalai Tesque, Inc. (Kyoto, Japan).

*ThT assays with heating*. All proteins were dissolved in deionized water, except $C_L$, Aβ40, TDP-43, and glucagon. For the exceptions, following buffers were used to make a complete dissolution: $C_L$, 10 mM sodium acetate buffer (pH 4.7); Aβ40, 0.05% (w/w) ammonia solution; TDP-43, 10 mM Tris-HCl buffer (pH 8.0) containing 4.0 M urea; glucagon, 20 mM HCl. Then, the pH was adjusted to those indicated in Table 1 with a small volume of 20 mM sodium phosphate buffer (pH 7.0) or 20 mM HCl. Protein concentrations were measured spectrophotometrically using individual theoretical molar extinction coefficients.

The assays of heating-dependent protein aggregation distinguishing amyloid fibrils and amorphous aggregates were carried out using a Hitachi F4500 fluorescence spectrophotometer (Tokyo, Japan) in the same manner as previously reported[16] and the equipment setup is shown in Supplementary Data Fig. 1.

*CD and TEM measurements*. Far-UV CD spectra (approximately 200–250 nm) were obtained by a model J-720 spectropolarimeter (Jasco Co., Ltd, Tokyo, Japan) at 25 °C using a quartz cell with a 1-mm path length. CD data were expressed as the mean residue ellipticity. To calculate the mean residue ellipticity, the total protein concentration of the sample solution was used. This obviously makes it difficult to convert to molar ellipticity when a large portion of the protein forms aggregates, as observed for glucagon and insulin with stirring (Fig. 3b). TEM images were obtained on a transmission electron microscope (H-7650, Hitachi High-Technologies Corporation) using 5 μL of 0.5% (w/v) hafnium chloride each as a staining agent.

*Seeding reactions*. Seeds were obtained from the preformed fibrils of heat-induced spontaneous amyloid formation under stirring and were moderately sonicated before seeding experiments. In the case of ThT assays, the conditions for sample preparation and measurements were the same as for standard experiments, except for the addition of 5% (v/v) seeds.

Calorimetric measurements were carried out using a MicroCal VP-DSC calorimeter (Malvern Panalytical, Ltd, Worcestershire, UK)[63]. The sample solution contained 0.5 mg/ml of each protein and the solution conditions were the same as those for the individual ThT assays with or without seeds. Sample and buffer solutions were carefully loaded into the DSC sample and reference cells, respectively, after being properly degassed in an evacuated chamber for 3 min at 25 °C. After the buffer–buffer baseline was subtracted from the sample data, apparent heat capacity (Cp) corresponding to the whole sample solution was evaluated using ORIGIN software (Microcal Inc.).

*Statistics and reproducibility*. Kinetic traces of three independent experiments are shown. When plate wells were used, the data are expressed as the mean ± s.e.m. for *n* = 5 wells. CD spectral and DSC measurements were performed twice.

**Reporting summary**. Further information on research design is available in the Nature Research Reporting Summary linked to this article.

## Data availability

The reporting summary for this article is available as a Supplementary file. The source data underlying Figs. 1a, b, 2a, c, e, d, 3a, b, 4a, b, and Supplementary Figs 2a, 3c, 4, 5a, c, 6a, c, 7a, 8a, b, 9a, 10 are provided in Supplemental Data 1. Any other remaining information is available from corresponding author upon reasonable request.

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

## Acknowledgements

This work was performed under the Cooperative Research Program for the Institute for Protein Research, Osaka University, CR-19-02, and was supported by Ministry of Education, Culture, Sports, Science and Technology (17H06352), Japan Society for the Promotion of Science (Core-to-Core Program A (Advanced Research Networks), and 20K06580) and SENTAN from the Japan Agency for Medical Research and Development, AMED. J.K. was supported by the National Research, Development and Innovation Fund of Hungary (K120391, 2017-1.2.1-NKP-2017-00002, and 2019-2.1.11-TÉT-2019-00079). D.E.O. was supported by the Lundbeck Foundation (R276-2018-671), the Novo Nordisk Foundation (NNF17OC0028806) and the Danish Research Council | Technology and Production (6111-00241B).

We thank Dr. Ruby Chen (Academia Sinica, Taipei, Taiwan) for kindly gifting the plasmid of TDP-43, Dr. Kenji Sasahara (IPR) for performing the calorimetric mea-surements, Pamina Kazman (Technische Universität München) for preparing V_L(PAT-1) and Ryan Bellman (UCL) for discussion.

## Author contributions

M.N. expressed and purified β2m and αSyn. Y.A.-O. and Y.H. expressed and purified C_L. T.S. E.C., and V.B. expressed and purified TTR. M.N. performed most of the experiments and analyzed the data together with Keiichi Yamaguchi, M.S. and Keisuke Yuzu. Y.K., K.I., H.M., V.B. supported the experiments and data analysis with αSyn and transthyretin. M.N., J.K., D.E.O., V.B., J.B., and Y.G. designed the study and wrote the manuscript with input from all co-authors.

## Competing interests

The authors declare no competing interests.
