## [Peer Review File · Communications Biology]

Reviewers' comments:

Reviewer #1 (Remarks to the Author):

The manuscript by Noji and co-workers presents analyses of conditions that trigger misfolding, conformational changes and amyloid formation in a variety of proteins under varying conditions of pH, temperature and concentration. The purported motivation is to characterize the proteins based on their relative tendencies to deviate from "Anfinsen's dogma". Based on ThT fluorescence, assays, circular dichroism, TEM and thermodynamic measurements, the authors classify the proteins into i) type 'S' or those that obey supersaturation dependent amyloidogenesis; ii) type 'A' or those that autonomously form amyloids; and iii) type 'B' or those that do not display a lag phase en route to amorphous aggregation at high temperature. The studies make important contributions to the field of protein folding vs. self-assembly and amyloidogenesis. However, for wider scope and greater effectivity, the manuscript must incorporate appropriate discussions of the following aspects:

1. It is unclear to the reader how mechanical agitation lowers the barrier associated with supersaturation? At the molecular level, how is it expected to affect nuclei formation and other pathways?
2. The distinction between the 'S' and the 'A' states seems somewhat arbitrary, even when taking into account length and net hydrophobicity. How does the charge distribution along the chain length, and other factors come into play? Greater elaboration of the differences would be appropriate.
3. The manuscript completely misses out any discussion on two related phenomena, namely i) liquid-liquid phase separation observed increasingly in disordered proteins, and ii) effect of extreme temperature and/or pressure on folding/misfolding and amyloidogenesis. The ramifications of the current findings on these emerging foci should be discussed. Recent articles that can be appropriately invoked include: Minton, J. Phys. Chem. B 2020, 124, 12; Ray et al, Nat. Chem 2020, 12, 705; Menon J. Phys. Chem. Lett. 2019, 10, 2453; Kozuch et al., 2019, J. Chem. Phys. 151, 185101; etc.

Minor points:

1. Terms like "rigidity of supersaturation" could be unclear to the general reader.
2. The individual heating-cooling plots in the ThT fluorescence data (eg. in Fig 1 a) are hard to discern and make sense of. I recommend using different panels.
3. There are several methods for estimation of protein conformational entropy, which itself varies with extant thermo-physical conditions. The authors should briefly elaborate on the reasons for selecting their chosen method, accompanied with a brief summary of the method in Supporting Information.
4. "Driving force of precipitation" used to describe the abscissa of the putative phase space in Fig 5a can be substituted with a better, more intuitive description. The fact that Fig 5a is a schematic should be mentioned in the legend of the plot.
5. Region IV in Fig 5b is supposed to represent 'B' type amorphous aggregates. However, the gray-blue assembly appears to contain ordered fibrils. This inconsistency should be clarified.

Reviewer #2 (Remarks to the Author):

Noji et al. probes the partitioning of proteins between folding and amyloid formation, and the dependence of amyloid formation on protein supersaturation. Authors classify the proteins into three classes depending on whether they aggregate upon heating without or with agitation. Agitation has been known for many decades to accelerate the protein aggregate formation. However, the classification of proteins into three distinct groups has not been done before. Authors correlate the aggregation behavior with average hydrophobicity of the protein, length, and the presence or absence of disulfide bonds. The manuscript is easy to read, and the data support the conclusions. This reviewer has the following concerns:

(1) Is there any correlation between the solubility limit of proteins at room temperatures and aggregation behavior at higher temperatures? Solubility limit of proteins can be quantitatively estimated using well established methods, for example, see Golovanov et al., JACS, 2004, 126, 8933-8939 and Kramer et al., Biophysical Journal, 2012, 102, 1907-1915. This is important because proteins never get heated in the real world.

(2) Aggregation propensity of proteins can be estimated using various methods that include AGGRESCAN, ZipperDB, TANGO, and others. Does this estimated aggregation propensity correlate with the nature of aggregation observed here with heating and agitation?

Reviewer #3 (Remarks to the Author):

The authors follow the aggregation of a variety of proteins that are known to aggregate or are have the potential to aggregate using as a main methodology the combination of light scattering (Ex/Em 445 nm) and ThT fluorescence (Ex445, Em485). The authors propose a supersaturation barrier that separates the denatured state from the aggregated state, and is dependent on the flexibility or rigidity of the denatured state of the proteins as determined by conformational entropy calculations. This a new claim as far as I know, and I don't disagree that there is a barrier that seems to be overcome in some cases more easily than others, and is certainly of interest to the folding field. Clearly, there is an effect of stirring on the potential for aggregation, and the authors use a magnetic stirrer set at 800 rpm(!) to induce aggregation. The authors propose three classes of proteins that have different potentials for aggregation - those that aggregate without stirring (B, which includes IAPP and insulin???) and type A and S which require stirring to promote aggregation. But, as it stands I am not convinced by the data that there is a barrier.

To make the case for a kinetic barrier, the kinetics of aggregation as a function of temperature needs to be measured (and if that only occurs with stirring then do that with stirring). In some cases there is a lag due to seeding, and in that case the kinetics will be more complicated but perhaps rather than LS just focus on the ThT fluorescence. To relate the barrier then to other proteins you would also need a dependence on the concentration (a plot of E_a versus concentration for each protein). In the data presented by the authors, only a single concentration is used (and the protein concentrations listed in Table 1 vary by a factor of 10!). To me there are no experiments that support the claim that different proteins have different barrier - maybe, but I just don't see the data to support it.

Another major problem is that the heating and cooling curves have been overlaid with ThT fluorescence and aggregation, with no indication of what is what, and are impossible to understand or interpret. The authors need to present the data in such a way that the reader has no problem

understanding which curve is due to heating and aggregation, or heating and ThT fluorescence, and so on.

The CD curves look fine, but its unclear if this is the CD of the aggregated material or whats left over in solution??? - whatever is aggregated is presumably not absorbing light. Further, if they are measuring aggregated material how is the concentration determined?? You can't present a molar ellipticity if the concentrations are unknown.

I'm also very concerned about the solvent conditions for the experiments - its mentioned that glucagon was solubilized in HCl, but the pH is listed as 7.0? Most of the proteins were dissolved in water which is great for CD experiments but the pH is listed as 7.0 - was that a measured pH or an assumed pH? If no buffer is present the results seem very difficult to interpret as pH effects could be contributing to the observed results.

Overall, I think the results are interesting and the idea of a supersaturation barrier that can be crossed more easily by some proteins than others is new. But, the barrier needs to be substantiated with more clear experiments. Also, as there is a relationship to stirring and crossing the barrier, a more thorough discussion of this in light of disease would be interesting.

Responses to the reviewers

Reviewer #1

General comments: The manuscript by Noji and co-workers presents analyses of conditions that trigger misfolding, conformational changes and amyloid formation in a variety of proteins under varying conditions of pH, temperature and concentration. The purported motivation is to characterize the proteins based on their relative tendencies to deviate from “Anfinsen’s dogma”. Based on ThT fluorescence, assays, circular dichroism, TEM and thermodynamic measurements, the authors classify the proteins into i) type ‘S’ or those that obey supersaturation dependent amyloidogenesis; ii) type ‘A’ or those that autonomously form amyloids; and iii) type ‘B’ or those that do not display a lag phase en route to amorphous aggregation at high temperature. The studies make important contributions to the field of protein folding vs. self-assembly and amyloidogenesis.

Our response: Thank you for your positive comments.

However, for wider scope and greater effectivity, the manuscript must incorporate appropriate discussions of the following aspects:

Major comment 1: It is unclear to the reader how mechanical agitation lowers the barrier associated with supersaturation? At the molecular level, how is it expected to affect nuclei formation and other pathways?

Our response: Since our first report of the ultrasonication-induced amyloid formation¹, we have been studying the molecular mechanism of agitation- or ultrasonication-dependent amyloid formation²⁻⁴. To address the effects of mechanical agitation, we added a following paragraph in the Discussion: Three types of transitions on a general phase diagram of aggregation (page 10).

“In terms of the phase diagram of conformational states (Fig. 5a), stirring or ultrasonication is a kinetic factor modifying the apparent phase diagram. It is likely that the boundary between the metastable and labile regions is shifted downward upon agitation, decreasing the barrier of supersaturation and inducing spontaneous amyloid formation². At a molecular level, we proposed a theoretical model of ultrasonication-induced amyloid formation³, where denatured monomers are captured on the bubble surface during its growth and highly condensed by subsequent bubble collapse, so that they are transiently exposed to high temperatures. Thus, the dual effects of local condensation and local heating contribute to dramatically enhance the nucleation reaction. We further suggested that the local condensation of the denatured proteins at the water-air interface forms seed-competent conformation⁴. Consistent with this, market suppression of amyloid nucleation of A β was shown under agitation without water-air interface⁵.”

Major comment 2: The distinction between the ‘S’ and the ‘A’ states seems somewhat arbitrary, even when taking into account length and net hydrophobicity. How does the charge distribution along the chain length, and other factors come into play? Greater elaboration of the differences would be appropriate.

Our response: Considering this and Comment 2 of Reviewer 2, we examined the relationship between transition types (i.e., S, A and B types) and various factors which might determine transition types. These include average hydrophobicity, CamSol, protein solubility, AGGRESCAN, and Tango AGG scores as well as net charge (Supplementary Table 1, Supplementary Figs. 12 and 13). We added the following sentences in “Discussion, Factors determining distinct amyloidogenic transitions”.

(Page 11) “To address the mechanism underlying the distinct amyloidogenic transitions, we examined the relationship between transition types (i.e., S, A and B types) and various factors which might determine the types. Although determining the solubility of denatured proteins experimentally at high temperatures will be important, it is impractical at this stage. Thus, we examined factors related to solubility although they are relative values estimated at ambient temperatures. These include average hydrophobicity score⁶, CamSol score⁶, protein solubility score⁷, AGGRESCAN score⁸, and Tango AGG score⁸ as well as net charge (**Supplementary Table 1**). For the proteins and peptides examined, the scores of various factors were plotted against the number of amino acid residues (**Fig. 5b, Supplementary Figs. 12**). The plots included our results using short amyloidogenic peptides obtained from full-length proteins, i.e. K3⁹ and OVA peptide (pOVA)¹⁰.”

(Page 13) “Although the estimation of conformational entropy in denatured state as well as that of disulfide bonds is not straightforward, we tentatively employed the methodology often used for the analysis of conformational stability of globular proteins¹¹⁻¹³. Assuming that the main chain contributes 21 J K⁻¹ per mole of residues¹¹, we estimated the intrinsic conformational entropies of the denatured state “without” disulfide bonds (ΔS_{conf}), which were plotted against various possible factors (**Table 1, Fig. 5c, Supplementary Table 1, Supplementary Fig. 13**). Again, hydrophobicity and AGGRESCAN scores were factors highly correlating with transition types, suggesting the importance of hydrophobicity-related interactions.”

Major comment 3: The manuscript completely misses out any discussion on two related phenomena, namely i) liquid-liquid phase separation observed increasingly in disordered proteins, and ii) effect of extreme temperature and/or pressure on folding/misfolding and amyloidogenesis. The ramification of the current findings on these emerging foci should be discussed. Recent articles that can be appropriately invoked include: Minton, J. Phys. Chem. B 2020, 124, 12; Ray et al, Nat. Chem 2020, 12, 705; Menon J. Phys. Chem. Lett. 2019, 10,

2453; Kozuch et al., 2019, J. Chem. Phys. 151, 185101; etc.

Our response: Considering the important comments, we added following sentences.

i) Liquid-liquid phase separation (page 13, the end of Discussion):

“Finally, one of the most important phenomena related to amyloid formation is liquid-liquid phase separation observed increasingly in disordered proteins^{14,15}. There are cases that the amyloid formation is preceded by the liquid-liquid phase separation. For an example, low-complexity domain of FUS protein formed the phase-separated droplets before formation of more stable amyloid fibrils¹⁶. The results are consistent with the Ostwald’s ripening rule of crystallization, in which morphologies of crystals change with time guided by their kinetic accessibilities and thermodynamic stabilities^{4,17,18}. We assume that the “macroscopic” phase diagram of conformational states as Fig. 5a will be also useful for understanding the liquid-liquid phase separation, where “microscopic” phase diagram might apply to each droplet system.”

ii) Effect of extreme temperature and/or pressure (page 15, Conclusion):

“In conclusion, our results provide another example that the effects of extreme conditions of temperature or pressure on folding/misfolding are important for clarifying the phenomena under physiological conditions¹⁴. The linkage of folding/misfolding and the law of mass action enable amyloid formation even under physiological conditions where only a low amount of unfolded precursor exists.”

Minor comment 1: Terms like “rigidity of supersaturation” could be unclear to the general reader.

Our response: Considering the comment, we rephrased “rigidity” by “persistence”.

Minor comment 2: The individual heating-cooling plots in the ThT fluorescence data (eg. in Fig 1 a) are hard to discern and make sense of. I recommend using different panels.

Our response: The similar concern was raised by Reviewer 3, comment 3. In Figures 1-4, we used different line types to distinguish the different measurements rather than using different panels.

Minor comment 3: There are several methods for estimation of protein conformational entropy, which itself varies with extant thermo-physical conditions. The authors should briefly elaborate on the reasons for selecting their chosen method, accompanied with a brief summary of the method in Supporting Information.

Our response: First of all, we understand that the estimation of protein conformational entropy as well as that of disulfide bonds is not straightforward even if there might be several methods. We employed the one often used for calorimetric study of protein folding/unfolding in which the main chain contributes 21 J K^{-1} per mole of

residues¹¹⁻¹³.”

We revised the sentence on page 13 as above: “Although the estimation of conformational entropy in denatured state as well as that of disulfide bonds is not straightforward, we tentatively employed the methodology often used for the analysis of conformational stability of globular proteins¹¹⁻¹³. Assuming that the main chain contributes 21 J K⁻¹ per mole of residues¹¹, we estimated the intrinsic conformational entropies of the denatured state “without” disulfide bonds (ΔS_{conf}), which were plotted against various possible factors (**Table 1, Fig. 5c, Supplementary Table 1, Supplementary Fig. 13**). Again, hydrophobicity and AGGRESCAN scores were factors highly correlating with transition types, suggesting the importance of hydrophobicity-related interactions.”

Minor comment 4: “Driving force of precipitation” used to describe the abscissa of the putative phase space in Fig 5a can be substituted with a better, more intuitive description. The fact that Fig 5a is a schematic should be mentioned in the legend of the plot.

Our response: Because we used the term “Driving force of precipitation” in our previous papers (e.g. ²), we would like to continue to use this terminology. Considering the comment, we added a sentence in the legend of Fig. 5a that “An increase in the salt concentration or temperature often increases the driving force of precipitation (abscissa), thus decreasing protein solubility (ordinate)”.

We also described that Fig 5a is a schematic phase diagram.

Minor comment 5: Region IV in Fig 5b is supposed to represent ‘B’ type amorphous aggregates. However, the gray-blue assembly appears to contain ordered fibrils. This inconsistency should be clarified.

Our response: In the original manuscript (page 9, Definition of *Type B proteins*), we already explained that “it is possible that their overall amorphous characteristics include amyloid cores (β -spines) producing a “‘uzzy coat’ morphology”, which we hope to clarify the reviewer’s concern.

Reviewer #2

General comments: Noji et al. probes the partitioning of proteins between folding and amyloid formation, and the dependence of amyloid formation on protein supersaturation. Authors classify the proteins into three classes depending on whether they aggregate upon heating without or with agitation. Agitation has been known for many decades to accelerate the protein aggregate formation. However, the classification of proteins into three distinct groups has not been done before. Authors correlate the aggregation behavior with average hydrophobicity of the protein, length, and the presence or absence of disulfide bonds. The

manuscript is easy to read, and the data support the conclusions.

Our response: Thank you for your positive comments.

This reviewer has the following concerns:

Comment 1: Is there any correlation between the solubility limit of proteins at room temperatures and aggregation behavior at higher temperatures? Solubility limit of proteins can be quantitatively estimated using well established methods, for example, see Golovanov et al., JACS, 2004, 126, 8933-8939 and Kramer et al., Biophysical Journal, 2012, 102, 1907-1915. This is important because proteins never get heated in the real world.

Our response: Theoretically, solubility is a thermodynamic property and the temperature dependence of intrinsic solubility of a denatured protein is defined by an equation shown in Supplementary Fig. 13 (see also Noji et al. 2019¹⁹ for detail). In the case of β 2-m, we determined the temperature dependence of solubility of the denatured state at pH 7, showing that it has a minimum at approximately 70°C. This results in the cold- and heat-depolymerization of amyloid fibrils. We described this general effect of temperature in Discussion (page 10, “Three types of transitions on a general phase diagram of aggregation”) of the original manuscript.

Considering the comment, we added a sentence (page 10): “We previously studied with β 2m at pH 7.0 the relationship between the temperature dependence of solubility and amyloid formation, showing both of lowest solubility and maximal amyloid formation occurred at approximately 70 °C¹⁹.”

The reviewer also listed reference papers for the solubility measurements. However, it is impractical to measure experimentally the solubility at room temperature or high temperature for various proteins used. Considering this and other comments, we plotted the solubility scores (i.e. CamSol and Protein-Sol scores) as well as other parameters against the length or denatured-state conformational entropy (Supplementary Table 1 and Supplementary Figs. 11 and 12).

Considering the comment, we added a sentence on page 11: “Factors determining distinct amyloidogenic transitions. “To address the mechanism underlying the distinct amyloidogenic transitions, we examined the relationship between transition types (i.e., S, A and B types) and various factors which might determine the types. Although determining the solubility of denatured proteins experimentally at high temperature will be important, it is impractical at this stage. Thus, we used factors related to solubility although they are relative values estimated at ambient temperatures.”

Comment 2: Aggregation propensity of proteins can be estimated using various methods that include AGGRESCAN, ZipperDB, TANGO, and others. Does this estimated aggregation

propensity correlate with the nature of aggregation observed here with heating and agitation? Our response: This comment is the same as Reviewer 1, Major comment 2. Considering the comments, we examined the relationship between transition types (i.e., S, A and B types) and various factors which might determine transition types. These include average hydrophobic, CamSol, protein solubility, AGGRESCAN, and Tango AGG scores as well as net charge (Supplementary Table 1, Supplementary Fig. 12, 13). We added the following sentences in “Discussion, Factors determining distinct amyloidogenic transitions”.

(Page 11) “To address the mechanism underlying the distinct amyloidogenic transitions, we examined the relationship between transition types (i.e., S, A and B types) and various factors which might determine the types. These include average hydrophobicity, CamSol⁶, protein solubility⁷, AGGRESCAN⁸, and Tango AGG²⁰ scores as well as net charge (Supplementary Table 1)”.

(Page 13) “Although the estimation of conformational entropy in the denatured state as well as the contribution of disulfide bonds are not straightforward, we tentatively employed the methodology often used for the analysis of conformational stability of globular proteins¹¹⁻¹³. Assuming that the main chain contributes 21 J K⁻¹ per mole of residues¹¹, we estimated the intrinsic conformational entropies of the denatured state “without” disulfide bonds (ΔS_{conf}), which were plotted against various possible factors (Table 1, Supplementary Table 1, Fig. 5c, Supplementary Fig. 13). Again, hydrophobicity and AGGRESCAN scores were factors highly correlating with transition types, suggesting the importance of hydrophobicity-related factors”.

Reviewer #3

General comments: The authors follow the aggregation of a variety of proteins that are known to aggregate or are have the potential to aggregate using as a main methodology the combination of light scattering (Ex/Em 445 nm) and ThT fluorescence (Ex445, Em485). The authors propose a supersaturation barrier that separates the denatured state from the aggregated state, and is dependent on the flexibility or rigidity of the denatured state of the proteins as determined by conformational entropy calculations. This a new claim as far as I know, and I don't disagree that there is a barrier that seems to be overcome in some cases more easily than others, and is certainly of interest to the folding field.

Clearly, there is an effect of stirring on the potential for aggregation, and the authors use a magnetic stirrer set at 800 rpm(!) to induce aggregation.

Our response: We used the same methodology before (e.g. ²) and we do not think the speed surprisingly high.

The authors propose three classes of proteins that have different potentials for

aggregation - those that aggregate without stirring (B, which includes IAPP and insulin???) and type A and S which require stirring to promote aggregation. But, as it stands I am not convinced by the data that there is a barrier.

Overall, I think the results are interesting and the idea of a supersaturation barrier that can be crossed more easily by some proteins than others is new. But, the barrier needs to be substantiated with more clear experiments. Also, as there is a relationship to stirring and crossing the barrier, a more thorough discussion of this in light of disease would be interesting.

Our response: We appreciate the careful review and important criticisms. We revised the manuscript considering the specific comments as described below.

Specific comment 1: To make the case for a kinetic barrier, the kinetics of aggregation as a function of temperature needs to be measured (and if that only occurs with stirring then do that with stirring). In some cases there is a lag due to seeding, and in that case the kinetics will be more complicated but perhaps rather than LS just focus on the ThT fluorescence. To relate the barrier then to other proteins you would also need a dependence on the concentration (a plot of E_a versus concentration for each protein). In the data presented by the authors, only a single concentration is used (and the protein concentrations listed in Table 1 vary by a factor of 10!). To me there are no experiments that support the claim that different proteins have different barrier - maybe, but I just don't see the data to support it. .

Our response:

(1) Temperature dependence: The comment is similar to Reviewer 2, comment 1, so please see our response to Reviewer 2, comment 1. As for $\beta 2m$, we already reported the temperature dependence of amyloid formation in detail¹⁹. The results showed that the amyloid formation is accelerated by an increase in temperature up to 90°C, consistent with the kinetic and thermodynamic mechanisms of amyloid formation, where hydrophobic interactions play an important role. Although the temperature dependence of other proteins used here might be interesting, we do not think additional experiments are essential in order to focus on classification of three types of amyloid formation.

(2) Protein concentration dependence: The comment raised by the reviewer is important. We performed the additional experiments with $\beta 2m$ and added a paragraph:

“Protein concentration dependence” on page 8: “Protein concentration dependence: Heating experiments were done at 0.1, 1.0, 2.0, and 10.0 mg/ml $\beta 2m$. With stirring, the kinetics of amyloid formation were similar at 0.1, 1.0 and 2.0 mg/ml $\beta 2m$ and, at 10 mg/ml $\beta 2m$, it was slightly accelerated. The results suggested that, at

low β 2m concentrations, the rate-limiting step was not an association of monomers but a process related to unfolding of the native state. At 10 mg/ml β 2m without stirring, amorphous aggregation occurred at temperatures above 60°C, indicating that the S-type transition changed to the B-type transition”.

We also discussed the results on page 13 (Discussion): “The driving forces also increase with an increase in protein concentration; therefore, the type of transition will change from S to A and B at higher protein concentrations. This change of transition type occurred for β 2m at pH 7.0, where the S-type transition at 0.1 mg/ml changed to the B-type transition at 10 mg/ml (Supplementary Fig. 10). This corresponds to a move along the x-axis in the phase diagram (Fig. 5a) and therefore automatically leads to a phase border crossing”.

Specific comment 2: Another major problem is that the heating and cooling curves have been overlayed with ThT fluorescence and aggregation, with no indication of what is what, and are impossible to understand or interpret. The authors need to present the data in such a way that the reader has no problem understanding which curve is due to heating and aggregation, or heating and ThT fluorescence, and so on.

Our response: This the same comment as Reviewer 1, Minor comment 2. In Figures 1-4, we used different line types to distinguish the different measurements rather than using different panels.

Specific comment 3: The CD curves look fine, but its unclear if this is the CD of the aggregated material or whats left over in solution??? - whatever is aggregated is presumably not absorbing light. Further, if they are measuring aggregated material how is the concentration determined?? You can't present a molar ellipticity if the concentrations are unknown.

Our response: Although the reviewer’s comment is important, it is practically impossible to estimate the amount of aggregated protein and subtract that to get reliable MRE values for the CD spectra. Conventionally, we and others estimate the “apparent” ellipticities using the original protein concentrations. From the observed spectra, ellipticities, turbidity etc. we judge whether the spectra are valid or not. Considering the comment, we revised the method sentence as below: “To calculate the mean residue ellipticity, the total protein concentration of the sample solution was used. This obviously makes it difficult to convert to molar ellipticity when a significant portion of the protein forms aggregates, as observed for glucagon and insulin with stirring (Fig. 3b)”.

Specific comment 4: I'm also very concerned about the solvent conditions for the

experiments - its mentioned that glucagon was solubilized in HCl, but the pH is listed as 7.0? Most of the proteins were dissolved in water which is great for CD experiments but the pH is listed as 7.0 - was that a measured pH or an assumed pH? If no buffer is present the results seem very difficult to interpret as pH effects could be contributing to the observed results.

Our response: Considering the concern of the reviewer, we revised the Methods section:

ThT assays with heating (page 15). “All proteins were dissolved in deionized water, except C_L, A β ₄₀, TDP-43, and glucagon. For those exceptions, the following buffers were used to achieve complete dissolution: C_L, 10 mM sodium acetate buffer (pH 4.7); A β ₄₀, 0.05% (w/w) ammonia solution; TDP-43, 10 mM Tris-HCl buffer (pH 8.0) containing 4.0 M urea; glucagon, 20 mM HCl. Then, the pH was adjusted to those indicated in Table 1 with a small volume of 20 mM sodium phosphate buffer (pH 7.0) or 20 mM HCl”.

References for the point to point reply

- 1 Ohhashi, Y., Kihara, M., Naiki, H. & Goto, Y. Ultrasonication-induced amyloid fibril formation of beta2-microglobulin. *J Biol Chem* **280**, 32843–32848 (2005).
- 2 Yoshimura, Y., Lin, Y., Yagi, H., Lee, Y. H., Kitayama, H., Sakurai, K., So, M., Ogi, H., Naiki, H. & Goto, Y. Distinguishing crystal-like amyloid fibrils and glass-like amorphous aggregates from their kinetics of formation. *Proc Natl Acad Sci U S A* **109**, 14446–14451 (2012).
- 3 Nakajima, K., Ogi, H., Adachi, K., Noi, K., Hirao, M., Yagi, H. & Goto, Y. Nucleus factory on cavitation bubble for amyloid beta fibril. *Sci Rep* **6**, 22015 (2016).
- 4 Nitani, A., Muta, H., Adachi, M., So, M., Sasahara, K., Sakurai, K., Chatani, E., Naoe, K., Ogi, H., Hall, D. & Goto, Y. Heparin-dependent aggregation of hen egg white lysozyme reveals two distinct mechanisms of amyloid fibrillation. *J Biol Chem* **292**, 21219–21230 (2017).
- 5 Morinaga, A., Hasegawa, K., Nomura, R., Ookoshi, T., Ozawa, D., Goto, Y., Yamada, M. & Naiki, H. Critical role of interfaces and agitation on the nucleation of Abeta amyloid fibrils at low concentrations of Abeta monomers. *Biochim Biophys Acta* **1804**, 986–995 (2010).
- 6 Sormanni, P., Aprile, F. A. & Vendruscolo, M. The CamSol method of rational design of protein mutants with enhanced solubility. *J Mol Biol* **427**, 478–490 (2015).
- 7 Hebditch, M., Carballo-Amador, M. A., Charonis, S., Curtis, R. & Warwicker, J. Protein-Sol: a web tool for predicting protein solubility from sequence. *Bioinformatics* **33**, 3098–3100 (2017).
- 8 Conchillo-Sole, O., de Groot, N. S., Aviles, F. X., Vendrell, J., Daura, X. & Ventura, S. AGGRESCAN: a server for the prediction and evaluation of “hot spots” of aggregation

- in polypeptides. *BMC Bioinformatics* **8**, 65 (2007).
- 9 Muta, H., So, M., Sakurai, K., Kardos, J., Naiki, H. & Goto, Y. Amyloid formation under complicated conditions in which β 2-microglobulin coexists with its proteolytic fragments. *Biochemistry* **58**, 4925–4934 (2019).
- 10 Noji, M., So, M., Yamaguchi, K., Hojo, H., Onda, M., Akazawa-Ogawa, Y., Hagihara, Y. & Goto, Y. Heat-induced aggregation of hen ovalbumin suggests a key factor responsible for serpin polymerization. *Biochemistry* **57**, 5415–5426 (2018).
- 11 Schellman, J. A. The stability of hydrogen-bonded peptide structures in aqueous solution. *C R Trav Lab Carlsberg Chim* **29**, 230–259 (1955).
- 12 Pace, C. N., Grimsley, G. R., Thomson, J. A. & Barnett, B. J. Conformational stability and activity of ribonuclease T1 with zero, one, and two intact disulfide bonds. *J Biol Chem* **263**, 11820–11825 (1988).
- 13 Makhatazde, G. I. & Privalov, P. L. Energetics of protein structure. *Adv Protein Chem* **47**, 307–425 (1995).
- 14 Cinar, H., Fetahaj, Z., Cinar, S., Vernon, R. M., Chan, H. S. & Winter, R. H. A. Temperature, Hydrostatic Pressure, and Osmolyte Effects on Liquid-Liquid Phase Separation in Protein Condensates: Physical Chemistry and Biological Implications. *Chemistry* **25**, 13049–13069 (2019).
- 15 Minton, A. P. Simple Calculation of Phase Diagrams for Liquid-Liquid Phase Separation in Solutions of Two Macromolecular Solute Species. *J Phys Chem B* **124**, 2363–2370 (2020).
- 16 Murray, D. T., Kato, M., Lin, Y., Thurber, K. R., Hung, I., McKnight, S. L. & Tycko, R. Structure of FUS Protein Fibrils and Its Relevance to Self-Assembly and Phase Separation of Low-Complexity Domains. *Cell* **171**, 615–+ (2017).
- 17 Levin, A., Mason, T. O., Adler-Abramovich, L., Buell, A. K., Meisl, G., Galvagnion, C., Bram, Y., Stratford, S. A., Dobson, C. M., Knowles, T. P. & Gazit, E. Ostwald's rule of stages governs structural transitions and morphology of dipeptide supramolecular polymers. *Nat Commun* **5**, 5219 (2014).
- 18 So, M., Hall, D. & Goto, Y. Revisiting supersaturation as a factor determining amyloid fibrillation. *Curr Opin Struct Biol* **36**, 32–39 (2016).
- 19 Noji, M., Sasahara, K., Yamaguchi, K., So, M., Sakurai, K., Kardos, J., Naiki, H. & Goto, Y. Heating during agitation of β 2-microglobulin reveals that supersaturation breakdown is required for amyloid fibril formation at neutral pH. *J Biol Chem* **294**, 15826–15835 (2019).
- 20 Rousseau, F., Schymkowitz, J. & Serrano, L. Protein aggregation and amyloidosis: confusion of the kinds? *Curr Opin Struct Biol* **16**, 118–126 (2006).

REVIEWERS' COMMENTS:

Reviewer #1 (Remarks to the Author):

The authors have incorporated modifications that answer most of my queries. However, they have not satisfactorily placed a discussion on thermodynamic (temperature and pressure) perturbations in the current context; the single line in the concluding paragraph is not adequate. This aspect is important as combined thermodynamic and mechanical effects can be expected to have profound implications in deviations from Anfinsen's dogma and possible onset of amyloidogenesis. It would be crucial to understand if the seemingly disparate perturbations act cumulatively or offset each other, and whether the combined effects are uniform across various sequences. An appropriate discussion in this context should expand the readability and interpretation of the present work and possibly trigger future studies. The references to invoke in the discussion must include:

-Mishra et al, Angew Chem Int Ed, 2008, 47, 6518

-ref 39

-Menon et al., J. Phys. Chem. Lett. 2019, 10, 2453

-Kozuch et al., J. Chem. Phys. 2019, 151, 185101.

Reviewer #2 (Remarks to the Author):

Authors have sufficiently addressed my earlier concerns.

Reviewer #3 (Remarks to the Author):

All of my concerns were addressed adequately, and I think its in good shape for publication. The figures are easier now to interpret, and I think my own familiarity (now) with previous studies by these authors (ref 19) makes the claim for a supersaturation barrier much more convincing.

minor comment: page 5, under stirring at pH 2.0 and 50_____,
I think its missing mM?

Responses to the reviewers' comments

Reviewer #1

The authors have incorporated modifications that answer most of my queries. However, they have not satisfactorily placed a discussion on thermodynamic (temperature and pressure) perturbations in the current context; the single line in the concluding paragraph is not adequate. This aspect is important as combined thermodynamic and mechanical effects can be expected to have profound implications in deviations from Anfinsen's dogma and possible onset of amyloidogenesis. It would be crucial to understand if the seemingly disparate perturbations act cumulatively or offset each other, and whether the combined effects are uniform across various sequences. An appropriate discussion in this context should expand the readability and interpretation of the present work and possibly trigger future studies. The references to invoke in the discussion must include:

- Mishra et al, Angew Chem Int Ed, 2008, 47, 6518
- Menon et al., J. Phys. Chem. Lett. 2019, 10, 2453
- Kozuch et al., J. Chem. Phys. 2019, 151, 185101.

Our response: Thank you for your additional comments. Considering your comments, we added sentences in conclusion: "Combined effects of thermodynamics (i.e., reversible unfolding) and kinetics (i.e., breaking supersaturation) have profound implications in deviations from Anfinsen's dogma and possible onset of amyloidoses. The seemingly disparate perturbations (i.e., reversible unfolding and breakdown of supersaturation) act cumulatively each other, which seems common to various proteins." We added the three references you commented in the manuscript.

Reviewer #2: Authors have sufficiently addressed my earlier concerns.

Our response: Thank you for your evaluation.

Reviewer #3: All of my concerns were addressed adequately, and I think its in good shape for publication. The figures are easier now to interpret, and I think my own familiarity (now) with previous studies by these authors (ref 19) makes the claim for a supersaturation barrier much more convincing.

Minor comment: page 5, under stirring at pH 2.0 and 50_____, I think its missing mM?

Our response: Thank you for your evaluation. "50??" should be "50 °C" (degree C).